# 3-D Multi-Component Reverse Time Migration Method for Tunnel Seismic Data

**DOI:** 10.3390/s21093244

**Published:** 2021-05-07

**Authors:** Peng Guan, Cuifa Shao, Yuyong Jiao, Guohua Zhang, Bin Li, Jie Zhou, Pei Huang

**Affiliations:** Faculty of Engineering, China University of Geosciences, Lumo Road 388, Wuhan 430074, China; p.guan@cug.edu.cn (P.G.); shaocuifa16@mails.ucas.edu.cn (C.S.); ghzhang@whrsm.ac.cn (G.Z.); libin@cug.edu.cn (B.L.); zhoujiecug@cug.edu.cn (J.Z.); huangpei0323@163.com (P.H.)

**Keywords:** multi-component, reverse time migration, three-dimensional, tunnel advance prediction

## Abstract

Migration imaging is a key step in tunnel seismic data processing. Due to the limitation of tunnel space, tunnel seismic data are small-quantity, multi-component, and have a small offset. Kirchhoff migration based on the ray theory is limited to the migration aperture and has low migration imaging accuracy. Kirchhoff migration can no longer meet the requirements of high-precision migration imaging. The reverse time migration (RTM) method is used to realize cross-correlation imaging by reverse-time recursion principle of the wave equation. The 3-D RTM method cannot only overcome the effect of small offset, but also realize multi-component data imaging, which is the most accurate migration method for tunnel seismic data. In this paper, we will study the 3-D RTM method for multi-component tunnel seismic data. Combined with the modeled data and the measured data, the imaging accuracy of the 3-D Kirchhoff migration and 3-D RTM is analyzed in detail. By comparing single-component and multi-component Kirchhoff migration and RTM profile, the advantages of the multi-component RTM method are summarized. Compared with the Kirchhoff migration method, the 3-D RTM method has the following advantages: (1) it can overcome the effect of small offset and expand the range of migration imaging; (2) multi-component data can be realized to improve the energy of anomalous interface; (3) it can make full use of multiple waves to realize migration imaging and improve the resolution of the anomalous interface. The modeled data and the measured data prove the advantages of the 3-D multi-component RTM method.

## 1. Introduction

The tunnel seismic advance prediction method is one of the most effective geophysical methods for the prediction of long-distance geological disasters in the tunnel. This can accurately predict the geological anomalous from 100 m to 150 m, including fault fracture zone, karst cave, soft rock, and water-rich zone. [1,2]. Migration imaging is one of the core technologies of tunnel seismic data processing. The narrow tunnel space limits the acquisition of seismic data in the tunnel, which gives the tunnel seismic data the characteristics of a small offset, multiple components, and a small data volume. Some scholars have studied the tunnel seismic data migration imaging method. Ashida et al. proposed the Kirchhoff migration method based on scattering superposition theory, which can achieve fast imaging but has low imaging accuracy [3]. Luth et al. proposed that the Kirchhoff migration operator is limited to the volume of the Fresnel zone, to improve the imaging accuracy of tunnel seismic data with small offset apertures [4]. Compared with the Kirchhoff migration method [5,6,7,8], RTM [9] utilizes the cross-correlation principle of forwarding continuation and reverse continuation to realize imaging, and the imaging area is not limited by the migration aperture. RTM is the most accurate migration method at present and is especially suitable for the migration imaging of tunnel seismic data with a small offset.

Many scholars have studied the ground RTM method, made breakthrough progress in imaging theory, imaging condition, and calculation speed, and achieved good results in practical applications.

Chang et al. proposed the excitation time imaging for the RTM method [10]. Mulder et al. found that iterative migration can reduce the migration illusion [11]. Yan et al. extended the cross-correlation imaging condition and produced elastic images for angle domain analysis [12]. Fletcher et al. realized acoustic RTM in the tilted transverse isotropy [13]. Multiple waves can enhance the illumination of the target body and reduce the influence of false anomalies. Liu et al. used multiple waves to improve the imaging accuracy of RTM [14]. Silvestrov et al. realized the post-stack RTM imaging [15]. High-performance computing methods such as GPU [16] and FPGA [17] are also used to improve the computational efficiency of RTM. Zha et al. applied the tunnel velocity model to realize RTM imaging of the 2-D elastic wave equation [18].

Although the 2-D RTM algorithm has been applied to the migration imaging of tunnel seismic wavefield, the imaging accuracy still needs to be improved. Tunnel space is a 3-D space, and the data received by the geophone come from the whole 3-D space. The 2-D RTM algorithm cannot make full use of the vertical component data to achieve multi-component migration imaging. The 3-D RTM method can realize multi-component migration imaging simultaneously [19]. In theory, if an accurate velocity model can be provided, high-precision migration imaging of the targeted body can be achieved. Due to the small amount of seismic data in the tunnel, it is necessary to make full use of the limited data to realize the imaging of the targeted body in front of the tunnel face. To date, there have been few studies on 3-D tunnel seismic data migration imaging. Whether the 3-D RTM algorithm can achieve true 3-D imaging of the front of the tunnel face is still unknown, and it needs to be analyzed in detail.

In this paper, we will study 3-D multi-component RTM suitable for tunnel seismic data. The RTM algorithm is realized using a 3-D finite-difference time-domain equation. The 3-D modeled data demonstrate the advantages of the 3-D multi-component RTM method. The measured data show that, compared with the Kirchhoff migration, the 3-D multi-component RTM method can overcome the effect of small offset, and is more beneficial to the delineation of the range of anomalous bodies. Later tunnel excavation results also verify the accuracy of 3-D multi-component RTM imaging results.

## 2. 3-D RTM Imaging

### 2.1. Image-Forming Principle

RTM is a two-way wave migration method based on wave equation theory. According to the time consistency imaging principle, it is assumed that there are some positions in the ground at the reflection interface, where the arrival time of the downgoing wave (Figure 1a) is the same as that of the upgoing wave (Figure 1b). In the migration calculation, the received wave field is extended along the time axis from the last moment to the zero moments with the real underground velocity.

In the spatial wavefield at each moment, the points that conform to the imaging condition are selected, and then points are accumulated into a spatial profile to obtain the migration imaging results. The implementation process realizes RTM imaging by the cross-correlation calculation of forwarding continuation data of the source point and reverse continuation of the receiving point at different moments (Figure 1). The detailed calculation process is shown in Figure 2, which is divided into three steps:

Forward continuation: the wavelet is used as the excitation source, the forward simulation calculates the maximum time and a snapshot of the wavefield each time is saved;Reverse continuation: the three-component wavefield is used as the excitation source, and the reverse continuation calculation is carried out from the maximum time to zero time. The forward continuation wavefield as the corresponding time is extracted for the multiplication operation. The calculation results at all times are stacked;Imaging: RTM imaging of all single-shot data is completed according to the process in steps (1) and (2). Then, a Laplace filter is used to remove the low-frequency noise, and the migration imaging results are obtained.

### 2.2. Forward Modeling

Forward modeling is the key to RTM imaging. In this paper, a 3-D acoustic wave equation is used for forward modeling. The 3-D acoustic wave equation can be expressed in the form of the first-order stress velocity equations in the Cartesian coordinate
(1)$$\frac{\partial P}{\partial t}=-\rho v^{2}\left(\frac{\partial Vx}{\partial x}+\frac{\partial Vy}{\partial y}+\frac{\partial Vz}{\partial z}\right)$$
(2)$$\frac{\partial Vx}{\partial t}=-\frac{1}{\rho }\frac{\partial P}{\partial x}$$
(3)$$\frac{\partial Vy}{\partial t}=-\frac{1}{\rho }\frac{\partial P}{\partial y}$$
(4)$$\frac{\partial Vz}{\partial t}=-\frac{1}{\rho }\frac{\partial P}{\partial z}$$
where *P* is sound pressure, *v* is the velocity, *ρ* is the density. *Vx*, *Vy*, *Vz* are the vibration velocities in the *x*, *y*, and *z* directions, respectively. The first-order stress-velocity equations in 3-D space are solved by the staggered grid finite difference method of the second order in time and twelfth order in space. The perfectly matched layer is used as the boundary absorption condition.

## 3. Numerical Simulation

In this section, the 3-D multi-component RTM imaging of the modeled data is realized by numerical simulation, and the precision of the 3-D multi-component RTM imaging of the tunnel seismic data is analyzed in detail. As shown in Figure 3, karst cave and interlayer models are designed. The model size is 50 m × 200 m × 50 m. The uniform grid spacing is 0.5 m. The time step is 3.5 × 10^−5^ s. According to the conventional TSP geometry, two geophones (Figure 3, R-1, and R-2) are installed on the tunnel wall, and twenty-four shots are placed on one side of the tunnel wall (Figure 3 shots). The shot interval is 1.6 m. Ricker wavelet is selected as the source wavelet, and the peak frequency is 500 Hz [20].

### 3.1. Wave Field Analysis

According to the velocity model shown in Figure 3, the positions of geophones and shots are exchanged. Seismic waves are excited at the geophone R-1 and R-2 respectively, and twenty-four shot points receive seismic data. The snapshots of the wavefield at 20 ms and 40 ms moments are shown in Figure 4, which shows the propagation characteristics of acoustic waves in 3-D tunnel space. At 20 ms moment, the wavefront has not reached the cave, and there is only an incident P-wave, which propagates forward (Figure 4a). At 40 ms moment, the wavefront of the incident P-wave has reached the weak interlayer, and the reflection wave is generated at the interface between the cave and the interlayer (Figure 4b). As the wavefront continues to travel forward, the reflection wave creates multiple waves at the interface. Multiple waves make the wavefield more complicated.

Figure 5 shows the three-component wave field recording received by geophone R-1. Symbol A denotes the direct wave that has linear characteristics. The symbol A1 denotes the reflection wave from the tunnel-top interface and shows hyperbolic characteristics. The energy of the Z-component reflection event (A1) is the strongest. The symbol A denotes the direct wave, A1 the reflection wave from the tunnel top interface, B to D the reflection wave from the karst cave, E and F the reflection wave from the interlayer. G is the multiple waves and generated from the interlayer. The reflected waves of the X and Y component are very similar, but they have a slight difference from the Z-component. D and G are the most obvious.

### 3.2. Migration Imaging

In the TSP method, the fitting results of the first arrival time of the direct wave are often used as the migration velocity to realize migration imaging. The fitting results are used as the migration velocity, the single-component Kirchhoff migration, single-component RTM, multi-component Kirchhoff migration, and multi-component RTM imaging are achieved, respectively. The migration results are shown in Figure 6. The symbol A denotes the interferential wave. The symbols B, C, and D denote the reflection interfaces of the cave. The symbols E and F denote the reflection interfaces of the interlayer. The red circle denotes the karst cave. The light blue and red lines denote the interlayer. The red arrows denote the false anomalous interfaces. Compared with the 3-D RTM profile (Figure 6b,d), the energy of the interference wave (A) is the strongest, and the resolution of the reflection interface (E and F) is the lowest in the Kirchhoff migration profile (Figure 6a,c). The main reason is that the multiple waves will reduce the resolution of the interface in the Kirchhoff migration profile, while the RTM method can use multiple waves to realize imaging. Compared with the single-component Kirchhoff migration profile (Figure 6a), the multi-component Kirchhoff migration method (Figure 6c) can improve the resolution of the reflection interface (F) and suppress the false artifacts (red arrows). Compared with the single-component RTM profile, the energy of the interface is stronger in the multi-component RTM profile, especially the interface F. The energy of the false anomalous interfaces (the red arrows) is the strongest in the Kirchhoff migration profiles (Figure 6a,c) and relatively weak in the RTM profiles (Figure 6b,d). The statistical results of the signal-to-noise ratio (SNR) of the migration data show that the SNR of the multi-component RTM profile is the highest (Table 1). The multi-component RTM method can make full use of the limited three-component seismic data to realize migration imaging and enhance the energy of the reflection interface. We validate the advantages of the 3-D multi-component RTM method with the comparison of using the single-component RTM method, the single-component Kirchhoff migration method, and the multi-component Kirchhoff method.

## 4. Engineering Application

The tunnel belongs to a water diversion tunnel in Chongqing. The transverse width of the tunnel is 3.6 m. The surrounding rock is composed of limestone, argillaceous limestone. There are karst caves in some areas. A giant karst cave has appeared in the excavated areas and is filled with yellow ooze. A small-scale mud inrush disaster has occurred, as shown in Figure 7. To find the possible karst cave in front of the tunnel face, the conventional TSP method is used to carry out advanced geological prediction. Twenty-four shots were designed. The shot interval is 1.5 m. The borehole depth is 2.0 m. Two three-component acceleration geophones were used to collect seismic data. The data sample interval is 41.7 ×10^−6^ s. The number of samples per trace is 4096.

Figure 8 and Figure 9 show the three-component wave field records of geophones R-1 and R-2 processed by AGC. The SNR of the raw data is high, and the consistency of the three-component data is better. In Figure 8 and Figure 9, the symbol A denotes the direct wave, B the surface wave, and C the acoustic wave. Spectrum analysis results show that the main frequency bandwidth within 0 ms and 60 ms is between 80 Hz and 800 Hz (Figure 10). The frequency bandwidth is taken as the parameters of band-pass filtering.

Despite the high SNR data, raw seismic data still have some problems, such as strong noise and multiple waves. To solve these problems, a series of data-processing methods has been applied, including trace equalization, band-pass filtering, shaping filtering, predictive deconvolution, F-K filtering, and migration imaging. The seismic data-processing procedure is shown in Figure 11. The processed P-wave data are shown in Figure 12 and Figure 13. The continuous reflective events with high amplitude are considered as the possible reflection waves in the front of the tunnel face (Figure 12 and Figure 13, the red lines).

The first chosen arrival travel time of the direct wave is used to calculate the surrounding rock velocity, and the P-wave velocity is 5802 m/s. The P-wave velocity is used as migration velocity, and the single-component Kirchhoff migration, single-component RTM, multi-component Kirchhoff migration, and multi-component RTM imaging are achieved, respectively, for the processed data in Figure 12 and Figure 13, and then the final results are obtained by stack processing. The migration results are shown in Figure 14. The symbol A indicates migration artifacts from the direct wave which have not been completely removed. The symbol B1 to F1 denotes the possible position of the cave in the Z-direction and B2 to F2 is in the X-direction. The interface B1 and E1 have similar mileage in the three migration profiles. Compared with the RTM profiles (Figure 14b,d), the energy of the direct wave artifact (A) is the strongest in the Kirchhoff migration profile (Figure 14a). The interface characteristics of the single-component Kirchhoff migration profile and RTM profile are almost identical in the X and Z directions (Figure 14a,b,(B1 to F1, B2 to F2)). Due to the addition of X- and Z-direction data, the three-component RTM profile can better distinguish the location of the anomalous interface and determine whether it comes from the X direction or the Z direction (Figure 14b,(B1 to F1, B2 to F2)). The energy difference at the anomalous interface is small, and there are more anomalous bodies in the Kirchhoff migration profile (Figure 14a), which will reduce the resolution of the true anomalous bodies. Compared with the single-component Kirchhoff migration profile (Figure 14a), the resolution of the anomalous interfaces is improved (E1 and E2) and energy of the anomalous interfaces is enhanced, especially for interface (B1, B2, F1, and F2) in the multi-component Kirchhoff migration profile (Figure 14c). In the single-component RTM profile (Figure 14b), the energy difference at the anomalous interfaces is great, and the interface of anomalous bodies is concentrated, so the mileage of the anomalous interface can be determined. The statistical results of the SNR of the migration data show that the SNR of the multi-component RTM profile is the highest (Table 2). Compared with other profiles, the resolution of the anomalous interface is the highest in the multi-component RTM profile (Figure 14d).

Figure 15 shows the geological interpretation map based on the results of tunnel excavation. The results of tunnel excavation show that there are karst caves at the mileage K0+660 and K0+710, while there are dense areas of fractures and joints at the mileage K0+760. Those anomalous bodies have good correspondence with the multi-component RTM profiles (Figure 14d,(B1,E1,F1)).

## 5. Discussion

The seismic wavefield received by the three-component geophone comes from the whole 3-D space. The traditional two-dimensional migration method assumes that the processed seismic data only come from the observation surface. If the geological environment is complicated and the lateral reflection wave is seriously disturbed, this assumption will cause a large error in the migration imaging. True 3-D migration imaging can suppress the interference of the lateral reflection wave and improve the resolution of migration imaging of anomalous bodies.

The Kirchhoff migration method based on ray theory has low requirements regarding the accuracy of the velocity model, obvious advantages in the stability and efficiency of the calculation results, and has achieved good results in the migration imaging processing of ground seismic data [21]. However, the ray theory migration method is limited to small aperture imaging. The characteristics of tunnel seismic data are their small amount and small offset. Due to the limitation of migration aperture, the Kirchhoff migration method based on the ray theory cannot distinguish the vertical and horizontal anomalous interfaces in the migration profile (Figure 14a). Moreover, due to the small energy difference in the interface, the real anomalous interfaces cannot be accurately determined, and the migration imaging accuracy of the anomalous body is insufficient.

The Fresnel volume migration method [22] can significantly improve the arc noise, but it cannot realize migration imaging by using multiple waves. The RTM method can effectively overcome the influence of small offset. The 3-D RTM imaging can be realized by the recursive calculation of wave equation, and the anomalous interfaces can be better distinguished in the vertical and horizontal direction (Figure 14b,c). Moreover, multiple waves can be used to improve the resolution of the anomalous interface. However, due to the small amount of tunnel seismic data, the multi-component RTM method is still unable to realize the 3-D space imaging of anomalous bodies, and unable to determine the attitude of rocks and spatial distribution characteristics of anomalous bodies.

Compared with the single-component RTM method, the multi-component RTM method can make full use of the limited tunnel seismic data to achieve 3-D migration imaging, enhance the energy of the real anomalous interface and suppress the interference interface. Both the synthetic data and the measured data prove the advantages of the 3-D three-component RTM method. The tunnel excavation results show that the strong impedance amplitude interfaces delineated by the three-component RTM profile are more consistent with the actual geological environment than the results of single-component migration.

## 6. Conclusions

This paper proposes a 3-D three-component RTM method for tunnel seismic data. Through the comparison of the modeled data and the measured data, we obtain the following conclusions:
Compared with the single-component migration method, the multi-component migration method can enhance the energy of the real anomalous interface;Compared with the Kirchhoff migration method, the RTM method can make full use of multiple waves to realize migration imaging, which will improve the resolution of the anomalous interfaces in front of the tunnel face;The RTM method can effectively overcome the problems of small offset and expand the range of migration imaging;Compared with single-component RTM, multi-component RTM can make full use of limited data to realize multi-component migration imaging, enhance the energy of anomalous interface and improve the resolution.

Migration imaging is the key to tunnel seismic data processing. The high-precision 3-D migration imaging method can improve the prediction accuracy and reduce the probability of false prediction. With the improvement in computer computing ability, the 3-D multi-component RTM method can be widely used in tunnel seismic data migration imaging. It has a great potential to improve the prediction accuracy and better serve the tunnel seismic prediction.

## Figures and Tables

**Figure 1 sensors-21-03244-f001:**
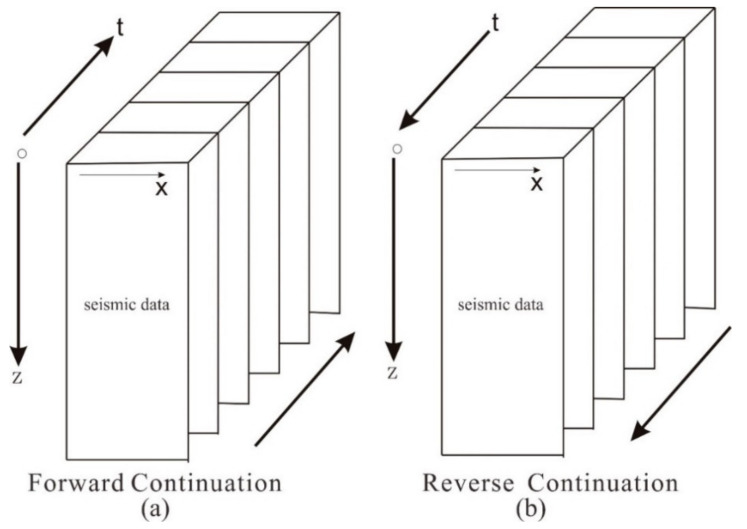
The principle of RTM imaging: (**a**) forwarding continuation; (**b**) reverse continuation.

**Figure 2 sensors-21-03244-f002:**
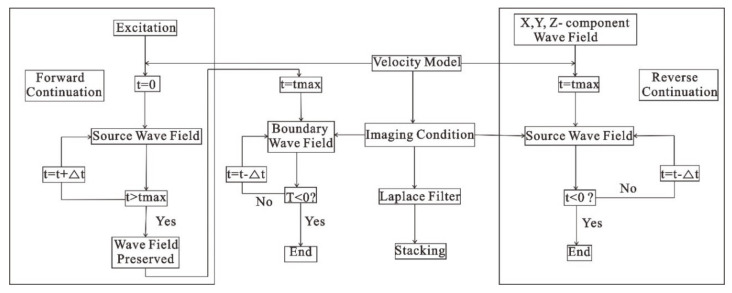
Multi-component RTM imaging calculation flow chart.

**Figure 3 sensors-21-03244-f003:**
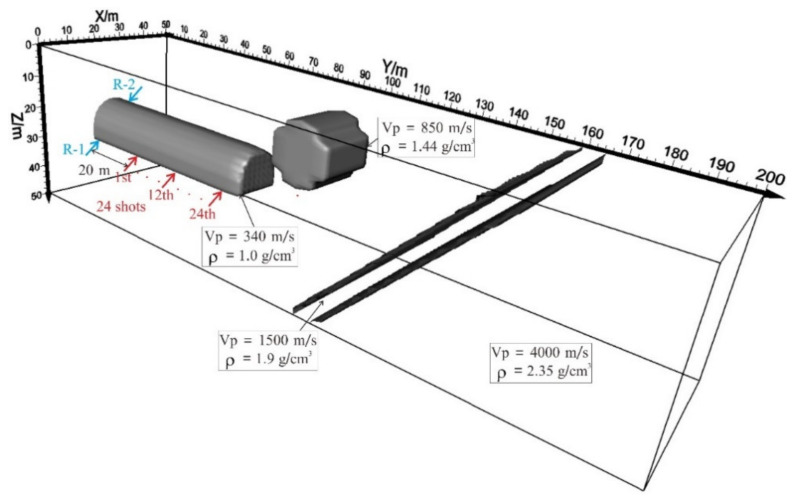
3-D tunnel velocity model. R-1 and R-2 are geophone points.

**Figure 4 sensors-21-03244-f004:**
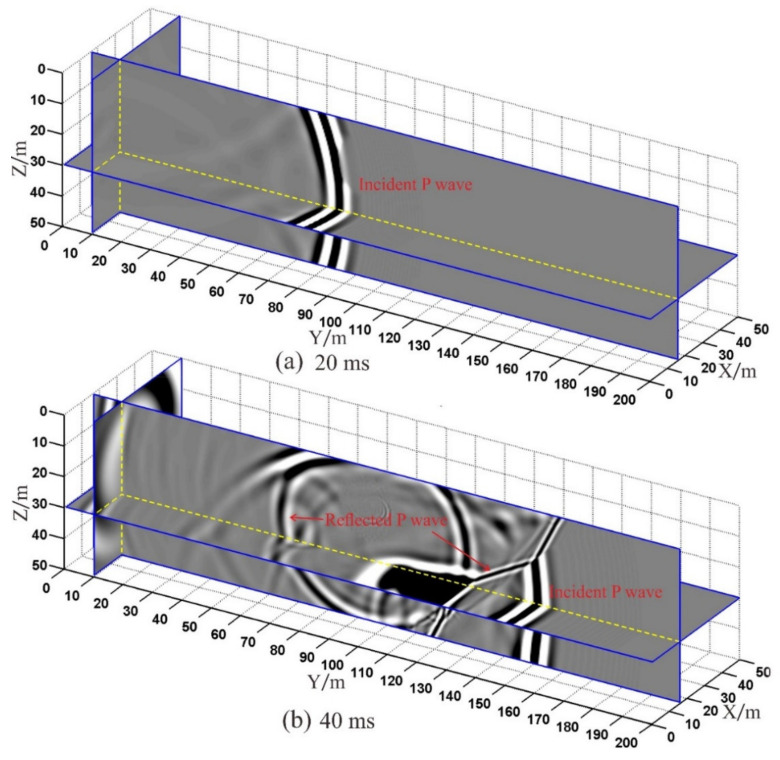
The snapshots of the wavefield: (**a**) 20 ms moment; (**b**) 40 ms moment.

**Figure 5 sensors-21-03244-f005:**
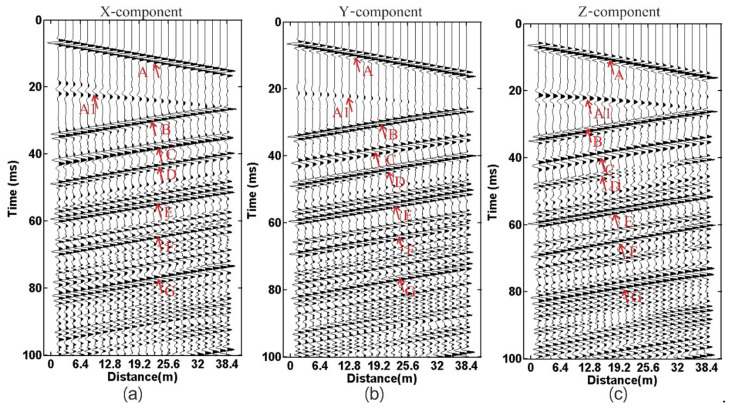
Wavefield recording: (**a**) X-component; (**b**) Y-component; (**c**) Z-component. The symbol A denotes the direct wave, A1 the reflection wave from the tunnel top interface, B to D the reflection wave from the karst cave, E and F the reflection wave from the interlayer, G the multiple waves.

**Figure 6 sensors-21-03244-f006:**
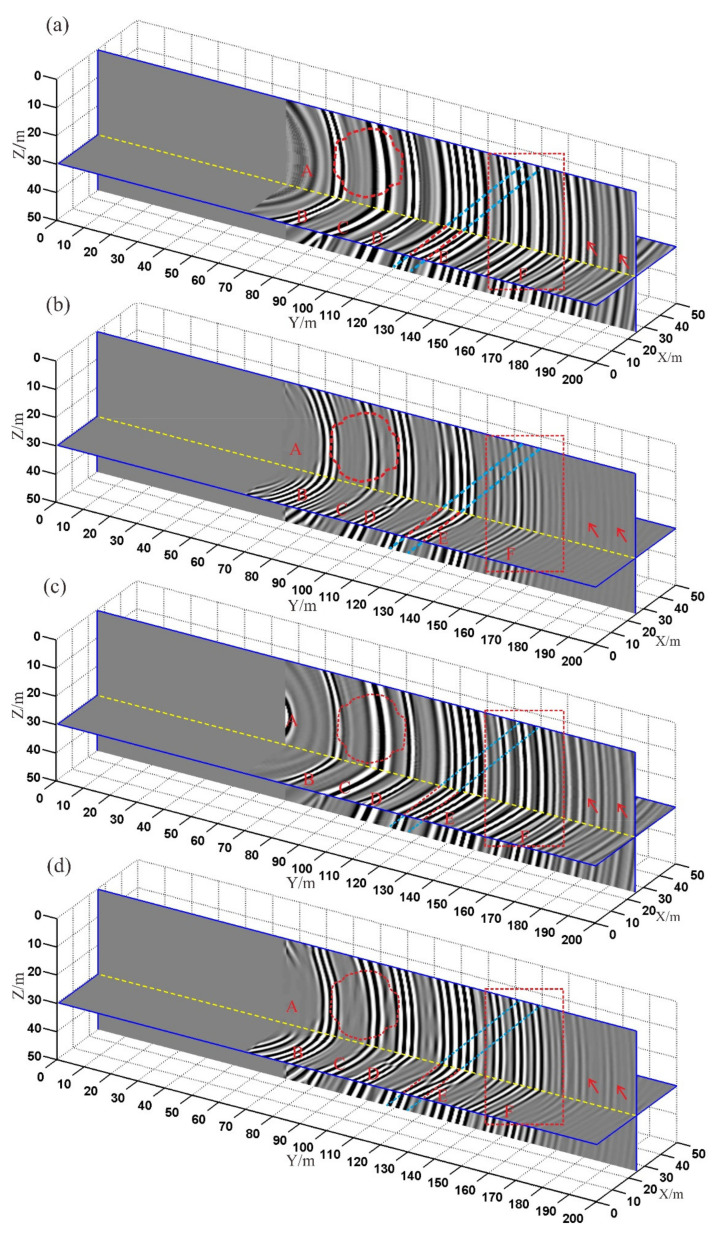
3-D migration profiles: (**a**) single-component Kirchhoff migration; (**b**) single-component RTM; (**c**) multi-component Kirchhoff migration; (**d**) multi-component RTM. The red circle denotes the karst cave. The light blue and red lines denote the interlayer. The red arrows denote the false anomalous interfaces.

**Figure 7 sensors-21-03244-f007:**
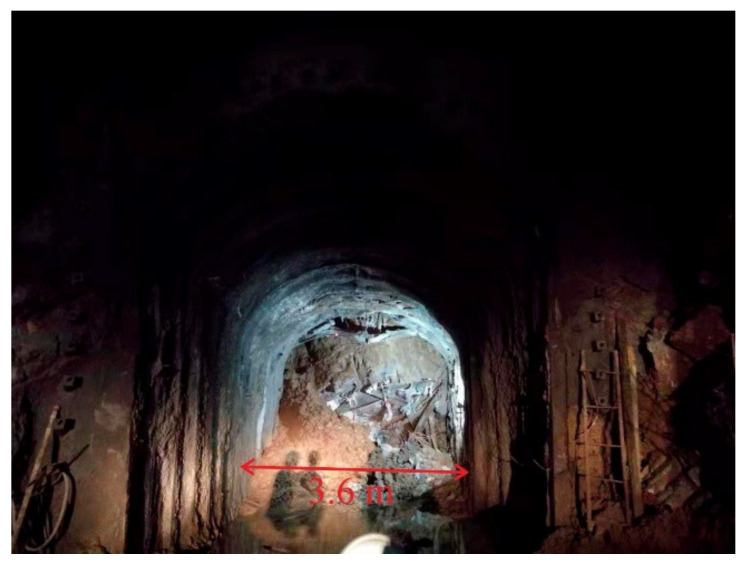
Photo picture of the tunnel.

**Figure 8 sensors-21-03244-f008:**
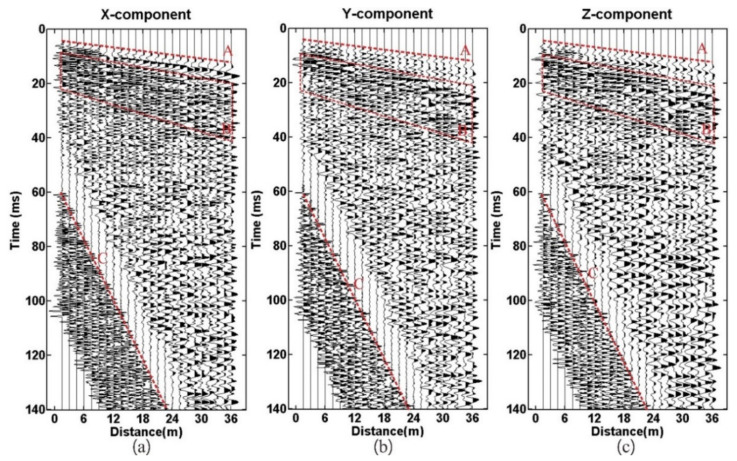
Geophone R-1: (**a**) X-component; (**b**) Y-component; (**c**) Z-component.

**Figure 9 sensors-21-03244-f009:**
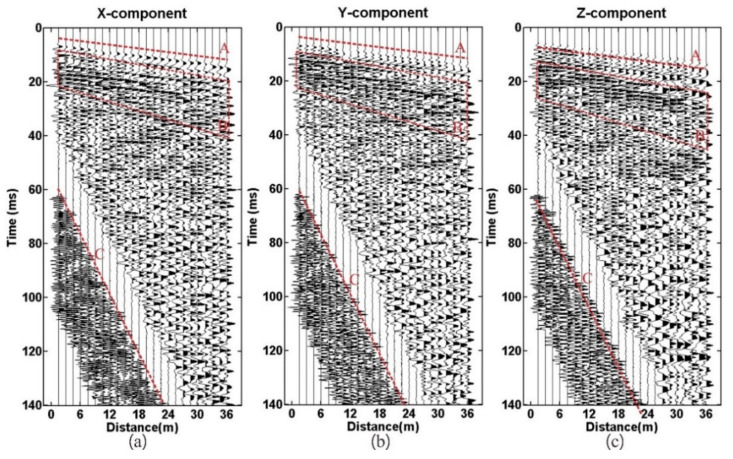
Geophone R-2: (**a**) X-component; (**b**) Y-component; (**c**) Z-component.

**Figure 10 sensors-21-03244-f010:**
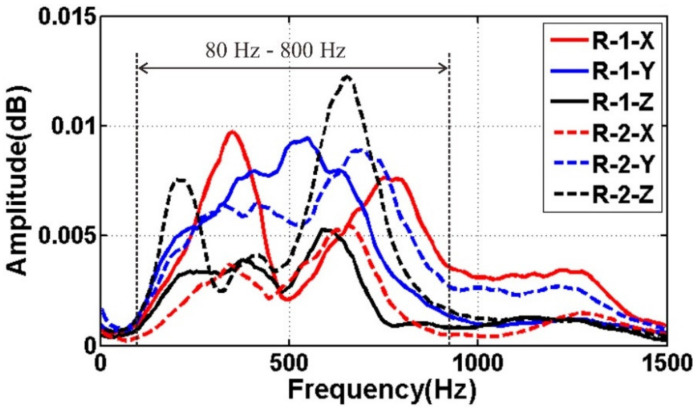
Spectrum analysis results.

**Figure 11 sensors-21-03244-f011:**
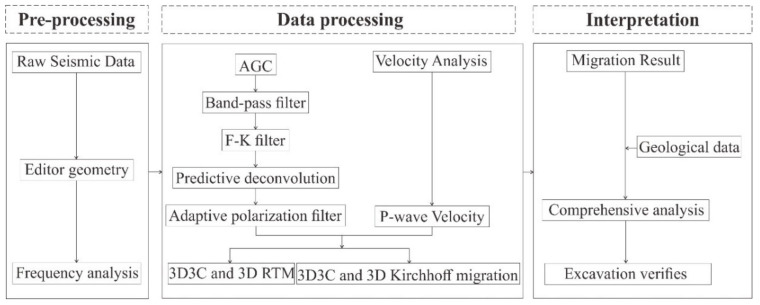
Flow chart of the seismic data processing. The 3D 3C and 3D RTM are 3-D multi-component P-wave RTM and single-component P-wave RTM, respectively. The 3D3C and 3D Kirchhoff migration are 3-D multi-component and single-component Kirchhoff migration, respectively.

**Figure 12 sensors-21-03244-f012:**
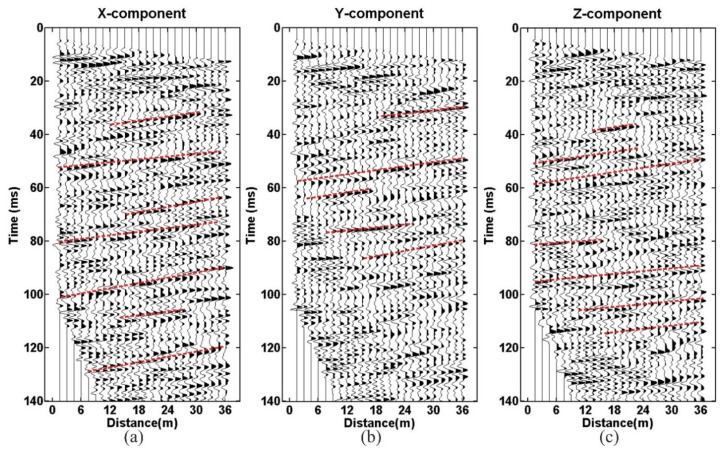
The processed data of geophone R-1: (**a**) X-component; (**b**) Y-component; (**c**) Z-component.

**Figure 13 sensors-21-03244-f013:**
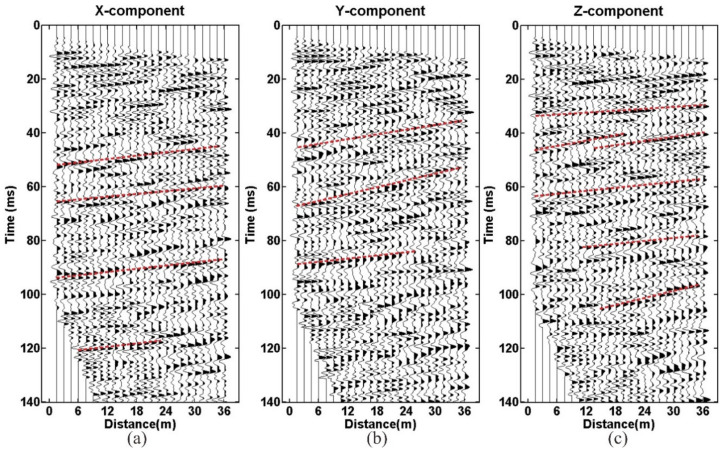
The processed data of geophone R-2: (**a**) X-component; (**b**) Y-component; (**c**) Z-component.

**Figure 14 sensors-21-03244-f014:**
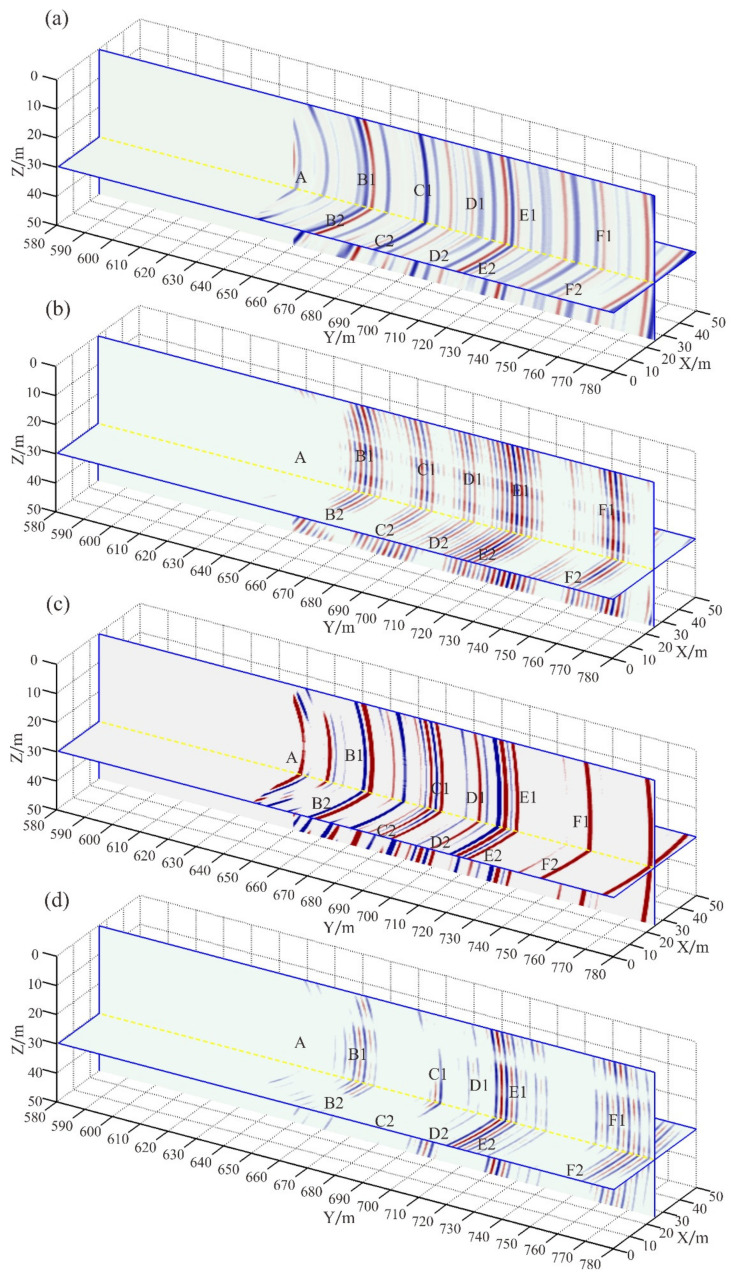
3-D migration profile: (**a**) single-component Kirchhoff migration profile; (**b**) single-component RTM profile; (**c**) multi-component Kirchhoff migration profile; (**d**) multi-component RTM profile.

**Figure 15 sensors-21-03244-f015:**
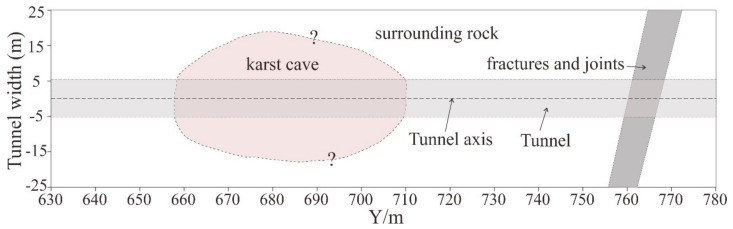
Geological interpretation map. The left and right extension boundary of the karst cave cannot be determined accurately.

**Table 1 sensors-21-03244-t001:** SNR of the migration results of the modeled data.

Profile	SNR
(a) Single-component Kirchhoff migration	169.48
(b) Single-component RTM	210.10
(c) Multi-component Kirchhoff migration	209.17
(d) Multi-component RTM	217.15

**Table 2 sensors-21-03244-t002:** SNR of the migration results of the measured data.

Profile	SNR
(a) Single-component Kirchhoff migration	279.10
(b) Single-component RTM	314.01
(c) Multi-component Kirchhoff migration	300.54
(d) Multi-component RTM	414.80

## Data Availability

The SEGY data used to support the findings of this study were supplied by P.G. under licence and so cannot be made freely available. Requests for access to these data should be made to corresponding author.

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
