# Peer review of "3-D Multi-Component Reverse Time Migration Method for Tunnel Seismic Data"

_sensors, 2021, doi:10.3390/s21093244_

Round 1
Reviewer 1 Report
In the paper entitled “3-D multi-component reverse time migration method for tunnel seismic data”, the 3-D RTM method is used for multi-component tunnel seismic data. Combined with the modelled data and the measured data, the imaging accuracy of the 3-D equal-plane migration and 3-D RTM is analyzed in detail. The modelled data and the measured data prove the advantages of the 3-D multi-component RTM method.
COMMENTS
The Introduction and the References should be enriched with recent relevant articles.
The Abstract should be better written.
LINE 237: 6. Conclusions (AND NOT Conclusion).
Author Response
In the paper entitled “3-D multi-component reverse time migration method for tunnel seismic data”, the 3-D RTM method is used for multi-component tunnel seismic data. Combined with the modeled data and the measured data, the imaging accuracy of the 3-D equal-plane migration and 3-D RTM is analyzed in detail. The modeled data and the measured data prove the advantages of the 3-D multi-component RTM method.
COMMENTS
- The Introduction and the References should be enriched with recent relevant articles.
Response:
Thank you for your comments. The introduction and the references have been rewritten and added the recent relevant articles.
The introduction is rewritten as
The tunnel seismic advance prediction method is one of the most effective geophysical methods for the prediction of long-distance geological disasters in the tunnel. It can accurately predict the geological anomalous 100 m to 150 m, including fault fracture zone, karst cave, soft rock, and water-rich zone, etc. [1,2]. Migration imaging is one of the core technologies of tunnel seismic data processing. The narrow tunnel space limits the acquisition of seismic data in the tunnel, which makes the tunnel seismic data have the characteristics of small offset, multi-component, and small data volume. Some scholars have studied the tunnel seismic data migration imaging method. Ashida et al [3] proposed the Kirchhoff migration method based on scattering superposition theory, which can achieve fast imaging but has low imaging accuracy. Luth et al [4] proposed that the Kirchhoff migration operator is limited to the volume of the Fresnel zone, to improve the imaging accuracy of tunnel seismic data with small offset apertures. Compared with the Kirchhoff migration method [5-8], RTM [9] utilizes the cross-correlation principle of forwarding continuation and reverse continuation to realize imaging, and the imaging area is not limited by the migration aperture. RTM is the most accurate migration method at present and is especially suitable for the migration imaging of tunnel seismic data with a small offset.
Many scholars have studied the ground RTM method, made breakthrough progress in imaging theory, imaging condition, and calculation speed, and achieved good results in practical application.
Chang et al [10] proposed the excitation time imaging for the RTM method. Mulder et al [11] found that iterative migration can reduce the migration illusion. Yan et al [12] extended the cross-correlation imaging condition and produced elastic images for angle domain analysis. Fletcher et al [13] realized acoustic RTM in the tilted transverse isotropy. Multiple waves can enhance the illumination of the target body and reduce the influence of false anomalies. Liu et al [14] used multiple waves to improve the imaging accuracy of RTM. Silvestrov et al [15] realized the post-stack RTM imaging. High-performance computing methods such as GPU [16] and FPGA [17] are also used to improve the computational efficiency of RTM. Zha et al [18] applied the tunnel velocity model to realize RTM imaging of the 2-D elastic wave equation.
Although the 2-D RTM algorithm has been applied to the migration imaging of tunnel seismic wavefield, the imaging accuracy still needs to be improved. Tunnel space is a 3-D space, and the data received by the geophone comes from the whole 3-D space. The 2-D RTM algorithm cannot make full use of the vertical component data to achieve multi-component migration imaging. 3-D RTM method can realize multi-component migration imaging simultaneously [19]. In theory, if an accurate velocity model can be provided, high-precision migration imaging of the targeted body can be achieved. Due to the small amount of seismic data in the tunnel, it is necessary to make full use of the limited data to realize the imaging of the targeted body in front of the tunnel face. Up to now, there are few studies on 3-D tunnel seismic data migration imaging. Whether the 3-D RTM algorithm can achieve true 3-D imaging of the front of the tunnel face is still unknown, and it needs to be analyzed in detail.
In this paper, we will study 3-D multi-component RTM suitable for tunnel seismic data. RTM algorithm is realized by using a 3-D finite-difference time-domain equation. The 3-D modeled data demonstrates the advantages of the 3-D multi-component RTM method. The measured data shows that, compared with the Kirchhoff migration, the 3-D multi-component RTM method can overcome the effect of small offset, and is more beneficial to delineate the range of anomalous bodies. Later tunnel excavation results also verify the accuracy of 3-D multi-component RTM imaging results.
The references are rewritten as
- Andisheh, A.; Ali, M.; Reza, N.; Mojtaba, Z.S.; Afshin, E. Prediction of geological hazardous zones in front of a tunnel face using TSP-203 and artificial neural networks. Tunnelling and Underground Space Technology. 2008, 23, 711-717.
- Shi, S.S.; Li, S. C.; Li, L.P.; Zhou, Z.Q.; Wang, J. Advance optimized classification and application of surrounding rock based on fuzzy analytic hierarchy process and Tunnel Seismic Prediction. Automation in Construction. 2014, 37, 217-222.
- Ashida, Y. Seismic imaging ahead of a tunnel face with three-component geophones. International Journal of Rock Mechanics and Mining Sciences. 2001, 38, 823-831.
- Luth, S.; Buske, S.; Giese R.; Goertz A. Fresnel volume migration of multicomponent data. Geophysics. 2005, 70, 121-129.
- Lüth, S.; Giese, R.; Otto, P.; Krüger, K.; Mielitz, S.; Bohlen, T.; Dickmann, T. Seismic investigations of the Piora Basin using S-wave conversions at the tunnel face of the Piora adit (Gotthard Base Tunnel). J. Rock Mech. Min. Sci. 2008, 45(1): 86-93.
- Rechlin, A.J.; Lüth, S.; Giese, R. OnSITE: integrated seismic imaging and interpretation for tunnel excavation. Proceedings of the International Conference on Rock Joints and Jointed Rock Masses. 2009,1-7.
- Tzavaras, J.; Buske, S.; Groß, K.; Shapiro, S. Three-dimensional seismic imaging of tunnels. J. Rock Mech. Min. Sci. 2012, 49, 12-20.
- Bellino, A.; Garibaldi, L.; Godio, A. An automatic method for data processing of seismic in tunneling. Journal of Applied Geophysics. 2013, 98, 243-253.
- Hu, Y.R.; McMechan, G.A. Imaging mining hazards within coalbeds using pre-stack wave equation migration of in-seam seismic survey data: A feasibility study with synthetic data. Journal of Applied Geophysics. 2007, 63, 24-34.
- Chang, W.F.; McMechan, G.A. Reverse-time migration of offset vertical seismic profiling data using the excitation-time imaging condition. Geophysics. 1986, 51(1), 67-84.
- Mulder, W.A.; Plessix, R.E. A comparison between one-way and two-way wave equation migration. Geophysics. 2004, 69 (6), 1491–
- Yan, J.; Sava, P. Isotropic angle-domain elastic reverse-time migration. 2008, 73 (6), S229–S239.
- Fletcher, R.P.; Du, X.; Fowler, P.J. Reverse time migration in tilted transversely isotropic (TTI) media. Geophysics. 2009, 74 (6), Wca179–Wca187.
- Liu, Y.K.; Hu, H.; Xie, X.B.; Zheng, Y.C.; Li, P. Reverse time migration of internal multiples for subsalt imaging. Geophysics. 2015, 80(5), S175–S185.
- Silvestrov, I.; Baina, R.; Landa, E. Poststack diffraction imaging using reverse-time migration. Geophysical Prospecting. 2016, 64, 129–142.
- Liu, H.W.; Li, B.; Liu, H.; Tong, X.L.; Liu, Q. The algorithm of high order finite difference pre-stack reverse time migration and GPU implementation. J. Geophys. 2010, 53(7), 1725-1733.
- Medeiros, V.; Barros, A.; Silva-Filho, A.; de Lima, M.E. High performance implementation of RTM seismic modeling on FPGAs: architecture, arithmetic and power issues. In: High-performance Computing Using FPGAs. 2013, Springer, pp. 305–334.
- Zha, X.J.; Gao, X.; Wang, W.; et al. Advanced prediction migration method research in tunnel engineering investigation. Chinese Journal of Geophysics (in Chinese). 2018, 61(3), 1150-1156.
- Gu, B.L.; Li, Z.Y.; Ma, X.N.; et al. Multi-component elastic reverse time migration based on the P- and S-wave separated velocity-stress equations. Journal of Applied of Geophysics. 2015, 112, 62-78.
- Li, H.; Zhu, P.; Ji, G.Z.; Zhang, Q. Modified image algorithm to simulate seismic channel waves in 3D tunnel model with rugged free surfaces. Geophysical Prospecting. 2016, 64, 1259-1274.
- Li, B.; Liu, G. F.; Liu, H. A Method of using GPU to accelerate seismic pre-stack time migration. Chinese Journal of Geophysics. 2009, 52(1), 242-249.
- Buske, S.; Gutjahr, S.; Christof, S. Fresnel volume migration of single-component seismic data. Geophysics, 2009, 74(6), WCA47–WCA55.
- The Abstract should be better written.
Response:
Thank you for your comments. The abstract is rewritten as
Migration imaging is a key step in tunnel seismic data processing. Due to the limitation of tunnel space, tunnel seismic data are small quantity, multi-component, and small offset. Kirchhoff migration based on the ray theory is limited to the migration aperture and has low migration imaging accuracy. Kirchhoff migration can no longer meet the requirements of high precision migration imaging. The reverse time migration (RTM) method is to realize cross-correlation imaging by reverse-time recursion principle of the wave equation. 3-D RTM method cannot only overcome the effect of small offset but also realize multi-component data imaging, which is the most accurate migration method for tunnel seismic data. In this paper, we will study the 3-D RTM method for multi-component tunnel seismic data. Combined with the modeled data and the measured data, the imaging accuracy of the 3-D Kirchhoff migration and 3-D RTM is analyzed in detail. By comparing single-component and multi-component Kirchhoff migration and RTM profile, the advantages of the multi-component RTM method are summarized. Compared with the Kirchhoff migration method, the 3-D RTM method has the following advantages: 1) it can overcome the effect of small offset and expand the range of migration imaging; 2) multi-component data can be realized to improve the energy of anomalous interface; 3) it can make full use of multiple waves to realize migration imaging and improve the resolution of the anomalous interface. The modeled data and the measured data prove the advantages of the 3-D multi-component RTM method.
- LINE 237: 6. Conclusions (AND NOT Conclusion).
Response:
Thank you for your comments. The “Conclusion” has been re-modified as “Conclusions”.
Reviewer 2 Report
This paper proposed the use of 3-D reverse time migration instead of 2-D reverse time migration for seismic data processing. It was shown through tests on synthetic and field data that the use of 3-D RTM improved the results visually over those from 2-D RTM. My main concern is that this improvement was mostly qualitative in nature and not quantified. The authors state that the new results have high signal-to-noise ratios without quantifying what these signal-to-noise ratios are. Especially in the case of synthetic tests, a RMSE or SNR figure can be given for 3-D RTM to compare it to 2-D RTM, etc. Otherwise, relying solely on visual quality can sometimes leads to incorrect conclusions.
Author Response
This paper proposed the use of 3-D reverse time migration instead of 2-D reverse time migration for seismic data processing. It was shown through tests on synthetic and field data that the use of 3-D RTM improved the results visually over those from 2-D RTM.
My main concern is that this improvement was mostly qualitative in nature and not quantified. The authors state that the new results have high signal-to-noise ratios without quantifying what these signal-to-noise ratios are. Especially in the case of synthetic tests, a RMSE or SNR figure can be given for 3-D RTM to compare it to 2-D RTM, etc. Otherwise, relying solely on visual quality can sometimes leads to incorrect conclusions.
Response:
Thank you for your good suggestions. For the synthetic data and the measured data, we calculated the signal-to-noise ratio (SNR) and make a quantitative evaluation.
Table 1 and Table 2 are shown the statistical results of the migration results of the modeled data and the measured data. The results show that the SNR of the multi-component RTM profile is the highest.
Reviewer 3 Report
My comments are reported in the attached file.

Author Response
- Line 127-128. The three-component reflection waves are very similar, but the multiple waves(H) have obvious differences.
Response:
Thank you for your comments. This part is rewritten as “The symbol A denotes the direct wave, A1 the reflection wave from the tunnel top interface, B to D the reflection wave from the karst cave, E and F the reflection wave from the interlayer. G is the multiple waves and generated from the interlayer. The reflected waves of the X and Y component are very similar, but they have a slight difference from the Z-component. In particular, D and G are the most obvious.”
- Line 160. “ 7 us.”
Response:
Thank you for your comments. This part is rewritten as “41.7 ×10-6 s”
- Line 177-178. including trace equalization, band-pass filtering, shaping filtering, predictive deconvolution, F-Kfiltering, and migration imaging.
Response:
Thank you for your comments. We added the flow chart of the seismic data processing. This part is rewritten as “To solve these problems, a series of data processing methods have been applied, including trace equalization, band-pass filtering, shaping filtering, predictive deconvolution, F-K filtering, and migration imaging. The seismic data processing procedure is shown in Figure 11.”
Reviewer 4 Report
Dear Authors,
Thank you for presenting this interesting approach for using the 3-component seismic wavefield in imaging ahead of a tunnel construction. The main purpose of the paper is to compare the new migration approach (Reverse Time Migration based on FD back-propagation of the seismic wavefield) with "conventional imaging approaches. I have a couple of comments, including some detailed suggestions and general remarks:
Details:
- Line 46: „data“ should be seismic wavefield, data is not received by geophones. They detect seismic waves.
- Line 76: wave filed → wave field
- Line 104: Sould the time step (3.5 e^(-5)) not be 3.5 * 10^(-5) s ?
- Line 119/129: The explanation of multiple waves is not satisfactory. One single interface would not produce multiples. Where are the multiples really generated? Between the void and the interface?
- Line 130, figure caption: explain here what is meant by letters A to H.
- Line 136: „Equal plane migration“ - What does this mean. I guess you mean the Kirchhoff type migration where amplitudes are distributed along the plane of equal two-way traveltime. Then you should use the appropriate term.
- Figures 3, 4, and 6: Model and wavefield snapshots as well as migration images should be plotted using the same orientation. Currently the model is rotated 180° versus the wavefield and migration image plots. In Figure 6, insert outlines of the structures for better comparison of the migration image with the „real“ structure.
- Line 165: „...the consistency of the three-component data is better.“ Better than what? Better needs a reference. Or you just want to say that the three component data is consistent (the three components are consistent with each other)?
- Lines 175-180: Processing including migration imaging – this is not correct. Processing should describe all steps applied before migration, then refer to data shown in Figures 11 and 12 which are not migrated data. Also, why do you claim that in Figs 11 and 12 „P-wave“ data is shown? You did not mention any processing step separating P from S waves (which is by the way not trivial if not even impossible under these conditions).
- Lines 185 and following: The terms „horizontal direction“ and „vertical direction“ are misleading. You are referring to horizontal and vertical sections from the 3D volume of migrated data. It is clear that the events (e.g. B1 and B2) are at the same positions, as they are representations of the same (3D) data.
- Line 190: Explain „interference wave“. This term is unusual. You may want to say that A indicates migration artefacts from the direct wave which has not been removed completely, is that correct?
- Figure 13: needs geological „ground truthing“ information. E.g., you could mark relevant geological features along the yellow line which should be running along the tunnel axis.
- Line 213: See line 46.
- Line 234 – 236: The phrase is a comparison: ...more consistent with the actual geological environment than the results of single component migration.
General remark: 3-C RTM is compared to single component RTM and single component classical migration, not considering the existence of 3-C Kirchhoff type migration (like the Fresnel Volume Migration cited in the introduction), which would probably be more appropriate here. State of art discussion is very short and is focusing on partly rather old publications, which may result from the fact that there has been little progress recently. Still, 3C-Properties of the data are considered in state-of the art migration techniques and should be compared with the method presented here.
Kind regards.
Author Response
Thank you for presenting this interesting approach for using the 3-component seismic wavefield in imaging ahead of tunnel construction. The main purpose of the paper is to compare the new migration approach (Reverse Time Migration based on FD back-propagation of the seismic wavefield) with "conventional imaging approaches. I have a couple of comments, including some detailed suggestions and general remarks:
Details:
- Line 46: „data“ should be seismic wavefield, data is not received by geophones. They detect seismic waves.
Response:
Thank you for your good suggestions. Data has been remodified as “seismic wavefield”.
- Line 76: wave filed → wave field
Response:
Thank you for your comments. The “wave filed” has been re-modified as “wavefield”.
- Line 104: Sould the time step (3.5 e^(-5)) not be 3.5 * 10^(-5) s ?
Response:
Thank you for your comments. The 3.5e-5 s is rewritten as 3.5×10-5 s.
- Line 119/129: The explanation of multiple waves is not satisfactory. One single interface would not produce multiples. Where are the multiples generated? Between the void and the interface?
Response:
Thank you for your comments. This part is rewritten as
Figure 5 shows the three-component wave field recording received by geophone R-1. Symbol A denotes the direct wave that has linear characteristics. The symbol A1 denotes the reflection wave from the tunnel top interface and shows hyperbolic characteristics. The energy of the Z-component reflection event (A1) is the strongest. The symbol A denotes the direct wave, A1 the reflection wave from the tunnel top interface, B to D the reflection wave from the karst cave, E and F the reflection wave from the interlayer. G is the multiple waves and generated from the interlayer. The reflected waves of the X and Y component are very similar, but they have a slight difference from the Z-component. In particular, D and G are most obvious.
- Line 130, figure caption: explain here what is meant by letters A to H.
Response:
Thank you for your comments. This part is rewritten as
The symbol A denotes the direct wave, A1 the reflection wave from the tunnel top interface, B to D the reflection wave from the karst cave, E and F the reflection wave from the interlayer, G the multiple waves.
- Line 136: „Equal plane migration“ - What does this mean. I guess you mean the Kirchhoff type migration where amplitudes are distributed along the plane of equal two-way traveltime. Then you should use the appropriate term.
Response:
Thank you for your good suggestions. Equal plane migration is a simplified version of the Kirchhoff migration method in the tunnel. To avoid misleading, the equal plane migration is re-modified as “Kirchhoff migration”.
- Figures 3, 4, and 6: Model and wavefield snapshots, as well as migration images, should be plotted using the same orientation. Currently, the model is rotated 180° versus the wavefield and migration image plots. In Figure 6, insert outlines of the structures for better comparison of the migration image with the “real” structure.
Response:
Thank you for your good suggestions. The model in Figure 3 is rotated 180°, so that the model, wavefield snapshots, and the migration images have the same angle, which is convenient for intuitive comparison.
The karst cave and interlayer model are added in Figure 6, which can visually show the corresponding relationship between the reflection interface and the actual model.
- Line 165: „...the consistency of the three-component data is better.“ Better than what? Better needs a reference. Or you just want to say that the three-component data is consistent (the three components are consistent with each other)?
Response:
Thank you for your comments. I want to emphasize that the three-component seismic data are consistent. By comparing the wavefields of the X, Y, and Z components, it is found that the data consistency of these three components is good.
- Lines 175-180: Processing including migration imaging – this is not correct. Processing should describe all steps applied before migration, then refer to data shown in Figures 11 and 12 which are not migrated data. Also, why do you claim that in Figs 11 and 12 „P-wave“ data is shown? You did not mention any processing step separating P from S waves (which is by the way not trivial if not even impossible under these conditions).
Response:
Thank you for your comments. In seismic data processing, the adaptive polarization filter method is used to extract P-wave data from the original data. To explain the seismic data processing procedure in detail, the flow chart of the seismic data processing is added in the paper.
- Lines 185 and following: The terms „horizontal direction“ and „vertical direction“ are misleading. You are referring to horizontal and vertical sections from the 3D volume of migrated data. The events (e.g. B1 and B2) are at the same positions, as they are representations of the same (3D) data.
Response:
Thank you for your comments. To avoid misleading, the horizontal direction and vertical direction are re-modified as X-direction and Z-direction, respectively.
- Line 190: Explain „interference wave“. This term is unusual. You may want to say that A indicates migration artifacts from the direct wave which has not been removed completely, is that correct?
Response:
Thank you for your good suggestions. You are right. A indicates migration artifacts from the direct wave which has not been removed completely.
- Figure 13: needs geological “ground truthing“ information. E.g., you could mark relevant geological features along the yellow line which should be running along the tunnel axis.
Response:
Thank you for your good suggestions. The geological interpretation map is added in the paper. According to the excavation results in the later period, the geological interpretation map on both sides of the central axis of the tunnel is drawn. Due to the lack of drilling data on the left and right sides of the tunnel, the extension distance of the left and right boundary of the karst cave cannot be accurately determined.
- Line 213: See line 46.
Response:
Thank you for your comments. Data has been remodified as “seismic wavefield”.
- Line 234 – 236: The phrase is a comparison: ...more consistent with the actual geological environment than the results of single-component migration.
Response:
Thank you for your good suggestions. This part is rewritten as “The tunnel excavation results show that the strong impedance amplitude interfaces delineated by the three-component RTM profile are more consistent with the actual geological environment than the results of single-component migration. ”
General remark: 3-C RTM is compared to single component RTM and single component classical migration, not considering the existence of 3-C Kirchhoff type migration (like the Fresnel Volume Migration cited in the introduction), which would probably be more appropriate here. State of art discussion is very short and is focusing on partly rather old publications, which may result from the fact that there has been little progress recently. Still, 3C-Properties of the data are considered in state-of-the-art migration techniques and should be compared with the method presented here.
Response:
Thank you for your good suggestions.
The three-component Kirchhoff migration is implemented in both the model data and the measured data and compared with the single-component and multi-component RTM methods. By comparing the advantages of the multi-component RTM method in tunnel seismic data migration imaging can be shown intuitively.
The discussion and conclusions are described again.
Discussion
The seismic wavefield received by the three-component geophone comes from the whole 3-D space. The traditional two-dimensional migration method assumes that the processed seismic data only comes from the observation surface. If the geological environment is complicated and the lateral reflection wave is seriously disturbed, this assumption will cause a large error in the migration imaging. True 3-D migration imaging can suppress the interference of the lateral reflection wave and improve the resolution of migration imaging of anomalous bodies.
The Kirchhoff migration method based on the ray theory has low requirements on the accuracy of the velocity model, has obvious advantages in the stability and efficiency of the calculation results, and has achieved good results in the migration imaging processing of ground seismic data [21]. However, the ray theory migration method is limited to small aperture imaging. The characteristics of tunnel seismic data are small amounts and small offset. Due to the limitation of migration aperture, the Kirchhoff migration method based on the ray theory cannot distinguish the vertical and horizontal anomalous interfaces in the migration profile (Figure 14(a)). Moreover, due to the small energy difference of the interface, the real anomalous interfaces cannot be accurately determined, and the migration imaging accuracy of the anomalous body is insufficient.
The Fresnel volume migration method [22] can significantly improve the arc noise, but it cannot realize migration imaging by using multiple waves. The RTM method can effectively overcome the influence of small offset. The 3-D RTM imaging can be realized by the recursive calculation of wave equation, and the anomalous interfaces can be better distinguished in the vertical and horizontal direction (Figure 14, (b) and (c)). Moreover, multiple waves can be used to improve the resolution of the anomalous interface. However, due to the small amount of tunnel seismic data, the multi-component RTM method is still unable to realize the 3-D space imaging of anomalous bodies, and unable to determine the attitude of rocks and spatial distribution characteristics of anomalous bodies.
Compared with the single-component RTM method, the multi-component RTM method can make full use of the limited tunnel seismic data to achieve 3-D migration imaging, enhance the energy of the real anomalous interface and suppress the interference interface. Both the synthetic data and the measured data prove the advantages of the 3-D three-component RTM method. The tunnel excavation results show that the strong impedance amplitude interfaces delineated by the three-component RTM profile are more consistent with the actual geological environment than the results of single-component migration.
Conclusions
This paper proposes a 3-D three-component RTM method for tunnel seismic data. Through the comparison of the modeled data and the measured data, we obtain the following conclusions:
- Compared with the single-component migration method, the multi-component migration method can enhance the energy of the real anomalous interface.
- Compared with the Kirchhoff migration method, the RTM method can make full use of multiple waves to realize migration imaging, which will improve the resolution of the anomalous interfaces in front of the tunnel face.
- The RTM method can effectively overcome the problems with small offset and expand the range of migration imaging.
- Compared with single-component RTM, multi-component RTM can make full use of limited data to realize multi-component migration imaging, enhance the energy of anomalous interface and improve the resolution.
Migration imaging is the key to tunnel seismic data processing. High precision 3-D migration imaging method can improve the prediction accuracy and reduce the probability of false prediction. With the improvement of computer computing ability, the 3-D multi-component RTM method can be widely used in tunnel seismic data migration imaging. It has a great significance to improve the prediction accuracy and better serve the tunnel seismic prediction.
Round 2
Reviewer 4 Report
Dear Authors,
congratulations for this quick revision of the manuscript. The quality of the paper has overall been improved and from my point of view there are only a few minor language editing issues which should be corrected before publication:
Page 3: "wave filed", please correct (wavefield)
Page 10: „interface wave“ should be „direct wave artifact“
Best regards.
Author Response
- Page 3: "wave filed", please correct (wavefield)
Response:
Thank you for your comments. The “wave filed” is re-modified as “wavefield”.
- Page 10: „interface wave“ should be „direct wave artifact“
Response:
Thank you for your comments. The “interface wave” is re-modified as “direct wave artifact.”